# Serological profiles of pan-coronavirus-specific responses in COVID-19 patients using a multiplexed electro-chemiluminescence-based testing platform

**Sidhartha Chaudhury[1], Jack Hutter[2], Jessica S. Bolton[3,4], Shilpa Hakre[5], Evelyn Mose[6], Amy Wooten[6], William O'Connell[6], Joseph Hudak[6], Shelly J. Krebs[5], Janice M. Darden[4], Jason A. Regules[3], Clinton K. Murray[7], Kayvon Modjarrad[5], Paul Scott[5], Sheila Peel[8], Elke S. Bergmann-Leitner**[3]*

**1** Center Enabling Capabilities, Walter Reed Army Institute of Research, Silver Spring, Maryland, United States of America, **2** Clinical Trials Center, Walter Reed Army Institute of Research, Silver Spring, Maryland, United States of America, **3** Immunology Core, Malaria Biologics Branch, Walter Reed Army Institute of Research, Silver Spring, Maryland, United States of America, **4** Henry M. Jackson Foundation for The Advancement of Military Medicine, Bethesda, MD, United States of America, **5** Emerging Infectious Diseases Branch, Walter Reed Army Institute of Research, Silver Spring, Maryland, United States of America, **6** Brian D. Allgood Army Community Hospital, Camp Humphreys, Pyeongtaek, South Korea, **7** United States Forces Korea Surgeon, Camp Humphreys, South Korea, **8** Diagnostics and Countermeasures Branch, Walter Reed Army Institute of Research, Silver Spring, Maryland, United States of America

* elke.s.bergmann-leitner.civ@mail.mil

**Data Availability Statement:** The serological data (expressed as Luminescence signal) are provided in S2 Table. These data are raw data and were log

## Abstract

Serological assessment of SARS-CoV-2 specific responses are an essential tool for determining the prevalence of past SARS-CoV-2 infections in the population especially when testing occurs after symptoms have developed and limited contact tracing is in place. The goal of our study was to test a new 10-plex electro-chemiluminescence-based assay to measure IgM and IgG responses to the spike proteins from multiple human coronaviruses including SARS-CoV-2, assess the epitope specificity of the SARS-CoV-2 antibody response against full-length spike protein, receptor-binding domain and N-terminal domain of the spike protein, and the nucleocapsid protein. We carried out the assay on samples collected from three sample groups: subjects diagnosed with COVID-19 from the U.S. Army hospital at Camp Humphreys in Pyeongtaek, South Korea; healthcare administrators from the same hospital but with no reported diagnosis of COVID-19; and pre-pandemic samples. We found that the new CoV-specific multiplex assay was highly sensitive allowing plasma samples to be diluted 1:30,000 with a robust signal. The reactivity of IgG responses to SARS-CoV-2 nucleocapsid protein and IgM responses to SARS-CoV-2 spike protein could distinguish COVID-19 samples from non-COVID-19 and pre-pandemic samples. The data from the three sample groups also revealed a unique pattern of cross-reactivity between SARS-CoV-2 and SARS-CoV-1, MERS-CoV, and seasonal coronaviruses HKU1 and OC43. Our findings show that the CoV-2 IgM response is highly specific while the CoV-2 IgG response is more cross-reactive across a range of human CoVs and also showed that IgM and IgG responses show distinct patterns of epitope specificity. In summary, this

transformed for the downstream analysis described in the manuscript.

**Funding:** The work was funded by the Military Infectious Disease Research program (MIDRP), which was not in the online database of funders. The funders did not have any influence on this study and the experimental plan.

**Competing interests:** The authors have declared that no competing interests exist.

multiplex assay was able to distinguish samples by COVID-19 status and characterize distinct trends in terms of cross-reactivity and fine-specificity in antibody responses, underscoring its potential value in diagnostic or serosurveillance efforts.

## Introduction

The current SARS-CoV-2 pandemic is unparalleled in recent history, and countries have developed diverse approaches to combat and manage transmission. Most countries are attempting to curtail the infection rates through strict social distancing rules, rigorous testing, and contact tracing of individuals potentially exposed to infectious individuals. A population-wide serological assessment of SARS-CoV-2 immunity would have several applications: (a) conduct surveillance to determine exposure rates, (b) investigate the feasibility of using antibody titers as markers of immunity, (c) examine the durability of SARS-CoV-2 antibody responses and protection from reinfection, (d) establish the serological landscape of pan-CoV antibody responses and determine whether pre-existing immunity to common human CoVs affect COVID-19 disease course; (e) screen individuals for participation in COVID-19 vaccine trials or for prioritization for receiving FDA-approved or emergency use authorized COVID-19 vaccines; and (f) screen for antibody reactivity to newly emerging SARS-CoV-2 strains.

Several tests have been developed for measuring SARS-CoV-2-specific IgG and IgM responses are currently available [1] and the data obtained from these tests suggest that SARS-CoV-2-specific antibody responses are reliably measurable by 2–3 weeks after onset of symptoms [2–4]. The presence of antibodies was detected in a minority of COVID-19 patients within one week of onset and seroconversion ranges from 90 to 100% by 15 days after onset [5, 6]. IgM seroconversion is seen around 12 days from onset, and IgG seroconversion around 13 days from onset [7], with the caveat that studies on antibody response dynamics have so far largely focused on convenience samples that may fail to capture the earliest emergence of these responses. However, recent findings suggest the possibility of a lack of durable antibody responses, whereby some percentage of individuals previously infected with SARS-CoV-2 that were seropositive following infection and later became seronegative during early convalescence, especially in cases where the SARS-CoV-2 infection resulted in asymptomatic infection or low disease severity [8, 9]. This finding agrees with previous studies in humans with other coronaviruses in that antibody-responses to CoV infections can be short-lived [10–12]. However, the durability of SARS-CoV-2 responses remains poorly characterized, and it is unclear how durability varies with respect to antibody fine-specificity, isotype profile, cross-reactivity to other coronaviruses, and how these aspects of the humoral immunity contribute to durable protection from reinfection.

Although numerous SARS-CoV-2-specific serological assays have been developed [13, 14], additional challenges remain in using these assays for surveillance or clinical management [15] including: 1) assay throughput, 2) the specificity of assay readouts to SARS-CoV-2 as compared to other related coronaviruses 3) and the need to perform sample testing at multiple dilutions due to the narrow linear range of the respective assay platform. Recently, we compared a newly developed multiplexed serological assay based on electro-chemiluminescence (ECLIA) customized for malaria serosurveillance with a qualified standard ELISA [16]. The results demonstrated superiority of the ECLIA-based assay in several aspects including a wide linear range eliminating the need for serial dilutions of test samples, low variability, robust reproducibility of the assay, and no signal reduction due to antigenic competition when testing

closely related antigens. Given that only 0.1 μL of sample is needed for the assay, and with the use of the pre-manufactured CoV antigen plates described here, five antigen plates (450 samples) could be run by a single operator in five hours, and with substantially higher throughput if using a liquid handler, this platform would be appropriate for use in high-throughput sero-surveillance applications.

The objective of the current study was to test the performance of a multiplexed pan-CoV ECLIA-based assay regarding its ability to establish a serological profile of responses to common beta CoVs (HKU-1, OC43), MERS-CoV, SARS-CoV-1 and SARS-CoV-2. For this study, three sample groups were compiled subjects diagnosed with COVID-19 from the Brian D. Allgood Army Community Hospital at Camp Humphreys in Pyeongtaek, South Korea from March 13 to April 18, 2020; healthcare administrators from the same hospital during the same time period but with no diagnosis of COVID-19; and pre-pandemic samples from several sources in the United States in 2019. The comparison of the COVID-19 and non-COVID-19 samples from the same site will enable us to compare antibody responses from COVID-19 diagnosed individuals from other individuals the same location and pandemic time period, who may have been exposed, but were not symptomatic or diagnosed with COVID-19. Comparison with pre-pandemic samples will determine whether there are differences in CoV exposure between pre-pandemic samples and pandemic samples, even in cases where there was no COVID-19 diagnosis. Our findings demonstrate the utility of this assay for SARS-CoV-2-sero-surveillance based on its high sensitivity and specificity as well as its ability to discern between pre-existing immunity to common human CoVs and SARS-CoV-2.

## Materials and methods

### Sample collection

For this analysis, plasma samples were obtained from a public health investigation of COVID-19 patients at the Brian D. Allgood Army Community Hospital (BDAACH) at Camp Humphreys in Pyeongtaek, South Korea from March 13 to April 18, 2020 (WRAIR#2755). Three sample groups were compiled: patients diagnosed with COVID-19; health care personnel assigned to the same hospital during the same time period whose duties did not include regular interaction with patients and who did not have a diagnosis of COVID-19 (these individuals were assumed to be unexposed or minimally exposed from living and working in an outbreak setting with a very low prevalence of COVID-19 at the time of collection); and pre-pandemic samples from several sources in the United States in 2019. COVID-19 and Control subjects were drawn from the same overall population: the U.S. Department of Defense military, civilian, and contractor population working at Camp Humphreys. The comparison of the COVID-19 and Control samples from the same site enabled us to measure antibody responses from COVID-19 diagnosed individuals from those in the same location and pandemic time period, who may have been exposed, but were not symptomatic or diagnosed with COVID-19. Samples from ten COVID-19 subjects and eight control subjects, matched by study location and population, were obtained and compared to a similar number of pre-pandemic samples (ten) using the multiplex ECLIA assay. As this was a retrospective analysis of COVID-19 samples collected during a public health investigation of a local outbreak, no *a priori* power calculation was carried out.

All COVID-19 diagnoses were confirmed using a nasopharyngeal swab and RT-PCR-based diagnostic assay (Centers for Disease Control 2019-nCoV RT-PCR diagnostic panel run on the Applied Biosystems 7500 platform). All Control subjects were also tested via nasopharyngeal swab and RT-PCR and confirmed to be negative for COVID-19 at the time of sample collection. COVID-19 disease severity was assessed as asymptomatic, mild (symptomatic but not

interfering with daily activity), moderate (interfering with daily activity but not requiring hospitalization), and severe (preventing daily activity and requiring hospitalization). All samples collected at BDAACH were sent to Walter Reed Army Institute of Research (WRAIR) for analysis. Pre-pandemic samples were obtained from a WRAIR blood collection protocol (WRAIR#2567) based on sample availability from August 2019 conducted in Silver Spring, Maryland. Finally, two pre-pandemic samples were commercially available as pooled plasma samples from GeminiBio (GemCell™ U.S. Origin Human Serum AB, Cat.No 100–512) that were delivered to WRAIR in 2018.

## Ethics approval and consent to participate

The plasma sample use was reviewed by the WRAIR Human Subjects' Protection Branch which determined that the research does not involve human subjects (NHSR protocol WRAIR #2567, WRAIR#2755, #EID-029) as the samples used were de-identified and no link between samples and subjects exists.

## Antigens

Antigens for this study were manufactured by MSD in a mammalian expression system (Expi 293 F) and printed onto the 10-plex plates by Meso Scale Diagnostics, LLC (Cat No K15362U (IgG), and K15363U (IgM), MSD, Rockville, Maryland). The antigens used were: HA-trimer Influenza A (Hong Kong H3), spike (soluble ectodomain with T4 trimerization domain) trimers for SARS-CoV-2, SARS-CoV-1, MERS-CoV, and betacoronaviruses HKU-1 and OC43, as well as the spike N-terminal domain (NTD, Q14-L303 of the SARS-CoV-2 spike sequence), receptor binding domain (RBD, R319-F541 of the SARS-CoV-2 spike sequence), and nucleocapsid protein (N; full length) for SARS-CoV-2, and bovine serum albumin (BSA).

## ECLIA

The MSD V-PLEX platform was used as 10-plex assays utilizing the pre-printed antigens described above with each printed on its own spot. Blocker A Solution (Cat.No R93BA, MSD) was added to the plates at 150 μl/well. The plates were sealed and incubated at room temperature (RT) for 1h on a plate shaker, shaking at 700 rpm. The plates were washed three times with 1x MSD Wash Buffer (Cat.No R61AA, 150 μl/well). Sera were diluted to 1:1000 dilution with Diluent 100 (Cat. No R50AA, MSD) and added to each well (50 μl/well). The same dilution was used for both IgM and IgG measurements. Plates were sealed and incubated at RT for 2h on a plate shaker, shaking at 700 rpm, then washed three times with 1x MSD Wash Buffer (150 μl/well). The detection antibody, SULFO-TAG either with anti-human IgG (Cat.No D20JL, MSD) antibody or anti-human IgM (Cat.No D20JP, MSD) was diluted to 2 μg/ml in Diluent 100 (MSD) and added to the wells (50 μl/well). The plates were sealed and incubated at RT for 1h on a plate shaker, shaking at 700 rpm. After washing, 150 μl a working solution of MSD GOLD Read Buffer B (Cat.No R60AM, MSD) was added to each well and immediately the plates were read on the MESO QuickPlex SQ 120 (MSD), per manufacturer's instructions. We assessed the dynamic range of the MSD V-PLEX platform using this antigen panel across a serial dilution range from 1:1000 to 1:30,000 and found high signal-to-noise ratio and a linear response across that entire span of concentrations (S1 Fig).

## Statistical analysis

The MSD assay provides a readout in units of mean luminescence intensity and all readouts were directly log-transformed prior to analysis without any normalization or subtraction of

background. Univariate analysis comparisons between groups (COVID-19, Control, and pre-COVID) were made using a Shapiro-Wilk Normality Test followed by a student's t test or a Wilcoxon signed rank test. We applied a multiple test correction using the Benjamin-Hochberg method; p-values were considered significant if their adjusted p-value was < 0.05. Principal Component Analysis (PCA) was carried out by normalizing and scaling the log-transformed values. Data points were colored by group, and ellipses were generated corresponding to 50% confidence intervals for each group, to identify general trends in the data set. Seropositivity for each CoV spike antigen for a given subject was assessed based on whether the readout for that antigen exceeded cutoff defined by the upper limit of the 99.9% confidence interval of the BSA (negative control) response, as determined from pooling the BSA response across all subjects in the study. This cutoff value was determined to be 8.85 for IgM and 8.96 for IgG in log-transformed units of mean luminescence intensity. Correlation plots were generated using pairwise Pearson correlation coefficients calculated from the log-transformed data. All statistical analysis was carried out in R using the *stats*, *ggplot2*, and *corrplot,*.

## Results

Table 1 provides a summary of the COVID-19 subjects, date of the first positive COVID-19 test, date of sample collection, disease severity and clinical indications as well as demographic information (age range and sex). All samples were collected within three weeks of the first positive COVID-19 test. One subject had severe COVID-19, six subjects had mild symptoms, and three subjects were asymptomatic. Demographic information on Control subjects is shown in S1 Table.

### SARS-CoV-2 specific IgG and IgM responses

Samples for all three groups were assayed on the MSD platform to determine IgM and IgG -specific responses to influenza H3, spike proteins for SARS-CoV-2, SARS-CoV-1, MERS-CoV, beta-coronaviruses OC43 and HKU1 relative to BSA (negative control) (summarized

**Table 1. Summary of COVID-19 subjects.**

| Subject ID | Age range (y. o.) [Sex] | Day of Symptom Onset* | Day of Sample Collection* | Disease Severity | Notes |
|---|---|---|---|---|---|
| i-0001 | 21–30 [M] | -1 | +17 | Mild | Chills, cough, from 18 to 15 days prior to participation (Day +17) |
| | | | +33 | | |
| | | | +37 | | |
| i-0002 | 21–30 [F] | 0 | +18 | Mild | Mild cough from 18 to 16 days prior to participation (Day +18) |
| i-0003 | 21–30 [F] | 0 | +1 | Mild | Anosmia, sore throat, cough and malaise 1 day prior to participation (Day +1), improved 9 days later |
| i-0004 | 51–60 [M] | 0 | +2 | Severe | Severe illness 2 days prior to participation (Day +2), hospitalized with hypoxia from -1 to 7 days after participation in the investigation; started improving by day 5; symptoms resolved by day 8 |
| | | | +7 | | |
| | | | +10 | | |
| i-0005 | 41–50 [M] | 0 | +3 | Mild | Mild cough 3 days prior to participation (Day +3), persisting through admission |
| i-0006 | 51–60 [M] | -12 | +4 | Mild | Chills, cough, runny nose, loss of appetite 16 days prior to participation (Day +4), persisted for weeks |
| i-0007 | 41–50 [M] | NA | +3 | Asymptomatic | never symptomatic |
| i-0008 | 41–50 [M] | NA | +2 | Asymptomatic | never symptomatic |
| i-0009 | 51–60 [F] | 0 | +2 | Mild | Mild cough 2 days prior to participation (Day +2), and persisted |
| i-0010 | 21–30 [F] | NA | +2 | Asymptomatic | never symptomatic |

* Day 0 defined as day of initial test positivity by nasopharyngeal COVID-19 test

in Fig 1). In terms of assay sensitivity, we found that COVID-19 samples had a roughly 300-fold higher IgM signal to SARS-CoV-2 spike protein and a 1000-fold higher IgG antibodies binding to SARS-CoV-2 spike antigen compared to BSA. In Control and pre-pandemic samples, there was no significant difference in IgM responses to SARS-CoV-2 spike protein and BSA, but the IgG response to the SARS-CoV-2 spike was approximately 10-fold higher ($p < 10^{-4}$) than to BSA, suggesting either some degree of cross-reactivity of pre-existing IgG antibodies to SARS-CoV-2 spike antigen in these samples or higher non-specific IgG binding to this antigen. We carried out serial dilutions to further assess assay sensitivity (S1 Fig).

We found that COVID-19 samples showed significant higher IgM responses to the SARS-COV-2 spike protein ($p < 10^{-7}$) than the control sample group and pre-pandemic group, while no differences between the three groups were noted for BSA or influenza. On average, COVID-19 samples showed 100- to 200-fold higher IgM responses to SARS-CoV-2 spike antigen than the control or pre-pandemic samples. For IgG responses, the COVID-19 samples also showed significantly higher responses than either the control or pre-pandemic samples ($p < 0.001$), while no differences were seen between the sample groups in terms of their reactivity to BSA or H3. Like the IgM responses, the IgG responses in the COVID-19 samples to SARS-CoV-2 spike protein were approximately 100- to 200-fold higher than in the control and pre-pandemic samples. In both IgM and IgG responses to CoV-2 spike protein, there was a single subject in the COVID-19 group, Subject i-0003, that was a low outlier, showing similar responses to Control and pre-pandemic samples. This subject had mild symptoms and their plasma sample was obtained only one day following initial test positivity; it is possible this was early in the infection course and the subject had not yet seroconverted.

## Cross-reactivity of SARS-CoV-2 antibody responses

Next, we analyzed differences in antibody binding response to the other CoV antigens between the three groups. COVID-19 subjects, in addition to showing significantly higher IgM binding

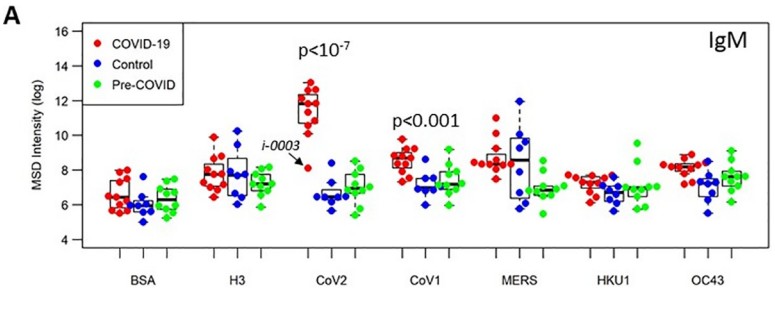

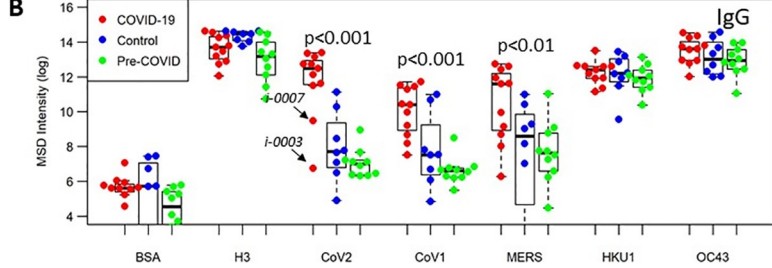

**Fig 1. Antibody responses to pan-CoV antigens.** Plasma levels of IgM (A) or IgG (B) from COVID-19 samples (n = 10, red), individuals with no known history of SARS-CoV-2 infection (Control group, n = 8, blue) and samples from US health donors (pre-COVID-19 group, n = 10, green) were tested at 1:1000 dilution; plate antigens shown are: BSA (negative contol); influenza H3 trimer (reference antigen), along with full-length spike proteins for CoVs: SARS-CoV-2, SARS-CoV-1, MERS-CoV, OC43 and HKU1.

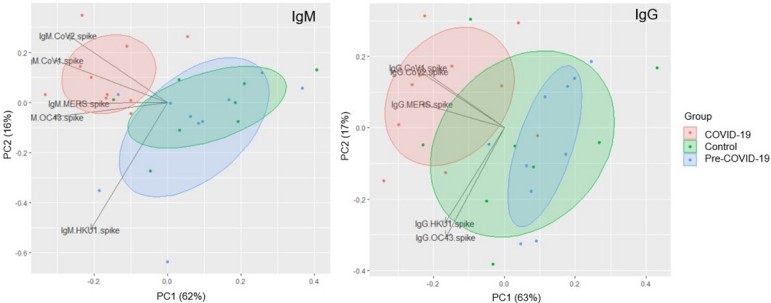

**Fig 2. PCA plot of IgM and IgG pan-CoV responses.** PCA plot showing IgM (left) and IgG (right) responses to spike proteins for SARS-CoV-2, SARS-CoV-1, MERS-CoV, OC43, and HKU1 for COVID-19 (red), Control (green), and pre-COVID (blue) samples. Each data point reflects a single sample colored by group; loading vectors reflecting the direction of the contribution of each parameter to the PCA plot is shown. Ellipses denote the 50% confidence interval for each group.

antibodies to SARS-CoV-2 spike compared to the Control and pre-COVID subjects, also showed significantly higher IgM binding antibodies to SARS-CoV-1 ($p < 0.001$), HKU1 ($p < 0.05$), and OC43 ($p < 0.01$) spike proteins (Fig 1A). This data suggests significant IgM cross-reactivity between SARS-CoV-2 and SARS CoV-1, HKU-1, and OC43. Likewise, we found that COVID-19 subjects, in addition to showing significantly higher IgG binding antibodies to SARS-CoV-2, also show significantly higher IgG binding antibodies to SARS-CoV-1 ($p < 0.001$) and MERS-CoV ($p < 0.01$) spike proteins, compared with Control and pre-COVID-19 subjects, suggesting that SARS-CoV-2 IgG antibodies may be cross-reactive with SARS-CoV-1 and MERS-CoV (Fig 1B).

To determine the distinguishing features between COVID-19 and non-COVID-19 samples, we generated a principal component analysis (PCA) plot of the IgM and IgG responses to these five CoV spike antigens (Fig 2). The PCA using IgM data demonstrates that samples display antibody responses largely along two major axes: SARS-CoV-1/SARS-CoV-2 vs. HKU1/ MERS-CoV. COVID-19 samples have high SARS-CoV-1/SARS-CoV-2 binding antibodies while Control and Pre-COVID-19 samples do not; HKU-1/OC43/MERS CoV responses appear to be independent of COVID-19 status. Analyzing the IgG responses revealed that the profile of cross-reactivity is different from that of IgM: samples show responses along two major axes: SARS-CoV-1/SARS-CoV-2/MERS-CoV and HKU-1/OC43. COVID-19 samples show high SARS-CoV-1/SARS-CoV-2/MERS-CoV responses, and responses along the HKU1/ OC43 axis appear to be independent of COVID-19 status.

## Seropositivity across CoV antigens

We next sought to determine the seropositivity of these samples across the five CoV spike antigens in the panel. We defined a cutoff above which a sample would be considered seropositive based on the IgG and IgM signal to BSA (noise). We set the cutoff at the upper limit of the 99.9% confidence interval calculated by pooling all the samples in the data set. This cutoff was determined to be 8.46 for IgM seropositivity and 8.96 for IgG seropositivity in log transformed units of luminescence intensity.

Overall, we found that 90% of the COVID-19 samples were seropositive for IgM binding antibodies to SARS-CoV-2, suggesting most COVID-19 patients tested here had seroconverted by the time the sample was taken (Fig 3, top). Additionally, 33% of CoV-2-seropositive COVID-19 samples were also seropositive for MERS-CoV, 33% were seropositive for both SARS-CoV-1, and 22% to both. By contrast, none of the Control and pre-pandemic subjects

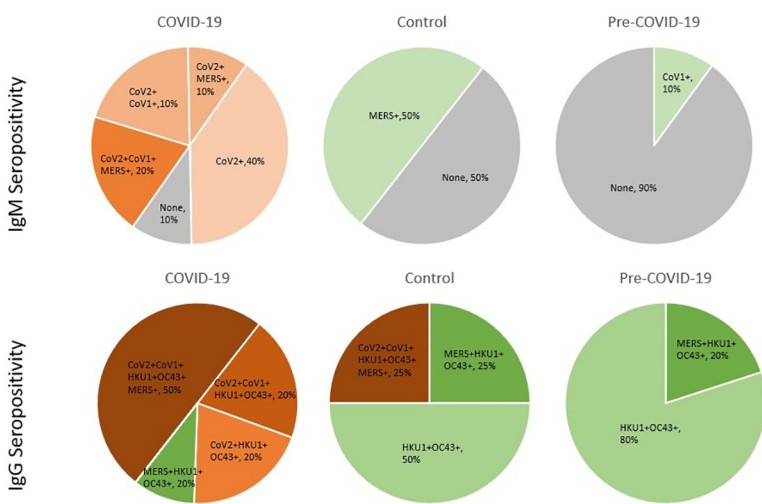

**Fig 3. Seropositivity Pan-CoV in IgM and IgG responses.** Pie charts showing percentage of subjects in the COVID-19 (left), Control (center), and Pre-COVID-19 (right) groups that are seropositive to each combination of CoV spike antigens for SARS-CoV-1, SARS-CoV-2, MERS-CoV, HKU1, and OC43, with '+' denoting seropositivity for the IgM (top) and IgG (bottom) responses. SARS-CoV-2 seropositivity is reflected by shades of orange, non-SARS-CoV-2 seropositivity is shown in shades of green, and no seropositivity to any CoV antigen is shown in gray.

were seropositive for SARS-CoV-2, and 50% of the Control and 90% of the pre-pandemic samples were seropositive to none of the CoV antigens, including OC43 and HKU1, again reflecting the lack of IgM response detected against these seasonal CoVs. Interestingly, 50% of Control samples were seropositive for MERS-CoV, compared to none of the pre-pandemic U. S. samples.

For the IgG responses, 90% of COVID-19 samples were seropositive for SARS-CoV-2, suggesting that one (of the 10) COVID-19 patients did not seroconvert at the time the sample was taken (Fig 3, bottom). Almost 60% of COVID-19 subjects that had seroconverted by IgG to SARS-CoV-2 were seropositive for all four other CoVs, suggesting substantial cross-reactivity in IgG SARS-CoV-2 binding antibodies. Interestingly, 25% of the subjects in the Control group showed seropositivity to not only SARS-CoV-2 but also to all five of the human CoVs tested, similar to what was observed in the COVID-19 group. A single subject in the COVID-19 group, aforementioned Subject i-0003, was seronegative to SARS-CoV-2 in both IgM and IgG responses. No clear association between seropositivity and symptom severity was observed in this data set.

None of the subjects in the pre-COVID-19 group showed seropositivity to SARS-CoV-2, but all showed seropositivity to the seasonal betacoronaviruses HKU1 and OC43. Overall two broad observations can be made from this seropositivity data: 1) IgM CoV binding antibodies likely reflect acute or recent infection while IgG CoV binding antibodies reflect both acute infection (in the case of SARS-CoV-2) or long-term memory responses (in the case of the seasonal CoVs) and 2) the IgG SARS-CoV-2 binding antibodies appear to be more cross-reactive than the IgM SARS-CoV-2 binding antibodies.

## Fine specificity of SARS-CoV-2 antibody responses

The multiplex assay contains additional antigenic targets of SARS-CoV-2, *i.e.*, RBD, NTD, and N (Fig 4). COVID-19 patients had significantly higher IgM levels directed at these antigens compared to the control groups. While all samples in the three groups had significant antibody responses to the seasonal CoVs (HKU-1, OC43), there was no significant recognition of the

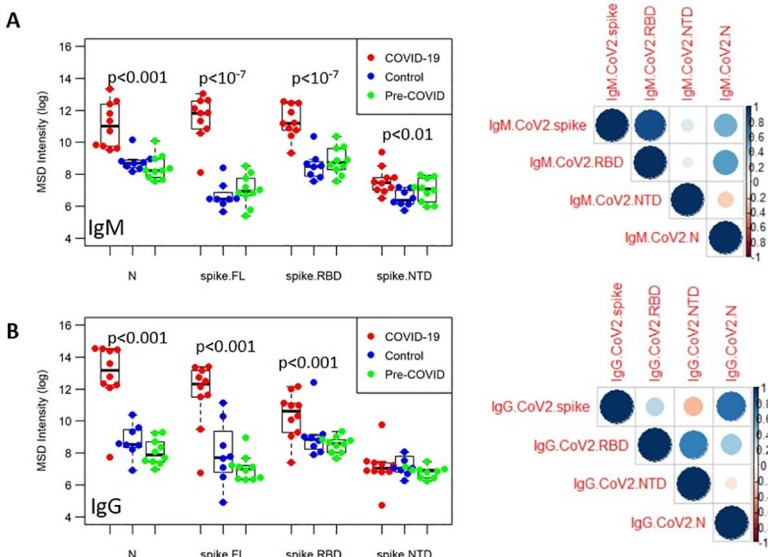

**Fig 4. Fine-specificity of SARS-CoV-2 specific antibody responses.** The IgM (A) and IgG (B) responses in samples from three groups (COVID-19, red; Control, blue; Pre-COVID, green) was assessed in the multiplex ECLIA platform against SARS-CoV-2 full-length spike protein, spike RBD, spike NTD, and nucleocapsid. Correlation matrices are shown on the right, with the color and size of the circles corresponding to pairwise Pearson correlation coefficient.

SARS-CoV-2 antigen fragments in the Control and pre-COVID-19 samples. The hierarchy of IgM binding to SARS-CoV-2 antigens reveals highest reactivity to spike and RBD, followed by binding to the nucleocapsid and the least reactivity to NTD. The antibody profile of SARS-CoV-2 specific IgG responses was different from the IgM profile. While COVID-19 patients had significantly higher IgG binding antibodies targeting the spike, RBD and nucleoprotein, the SARS-CoV-2-specific IgG responses to NTD were not significantly different between the three sample groups. As before, one subject in the COVID-19 group, aforementioned Subject i-0003, was a low outlier in responses to SARS-CoV-2 spike and RBD for both IgG and IgM responses.

To identify the relationship between the different antibody specificities, correlation matrices were generated for IgM and IgG responses (Fig 4) demonstrating that the magnitude of IgM SARS-CoV-2 spike binding antibodies correlated strongly with RBD responses. To a lesser extent, there was also a positive correlation between the nucleocapsid and RBD specific antibodies. The antibody profile of SARS-CoV-2-specific IgG was distinct from the IgM profiles as there was a weak correlation between nucleocapid and spike-specific responses and NTD with RBD specific antibodies, suggesting that the fine specificity between the IgM and IgG SARS-CoV-2 spike responses may differ, specifically that the IgM spike binding antibodies target epitopes largely to the RBD, while the IgG spike binding antibodies may be more focused on epitopes that include regions outside of the RBD itself, or target RBD epitopes unique to the whole-spike structure that are not recapitulated in the recombinant protein.

## Combining IgM and IgG CoV-2 responses to identify COVID-19 samples

We combined all the SARS-CoV-2 specific antigen readouts (full-length spike, RBD, NTD, and N) for IgM and IgG to determine if they could clearly distinguish COVID-19 samples from Control or Pre-COVID-19 samples. Using an unsupervised PCA approach, we show that these groups can be readily distinguished (Fig 5A), and that even a reduction from these 12

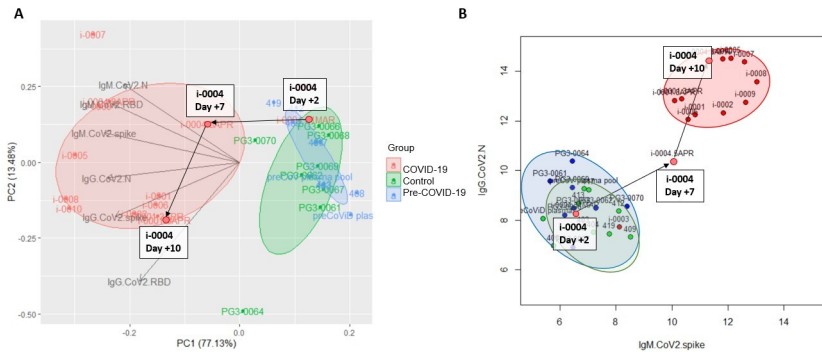

**Fig 5. Combining IgG and IgM responses to distinguish COVID-19 samples.** PCA plot of IgM and IgG responses to SARS-CoV-2 antigens spike, RBD, NTD, and Ncap (A) for COVID-19 (red), Control (green), and pre-COVID (blue) subjects. Loading vectors showing the direction of the contribution of each parameter to the PCA are shown. Ellipses correspond to 50% confidence intervals for each group. Scatterplot of IgM SARS-CoV-2 spike responses and IgG SARS-CoV-2 Ncap responses for all three groups. Longitudinal data for COVID-19 subject i-0004 collected on Day +2, Day +7, and Day +10, relative to day of initial test positivity, is highlighted as the subject seroconverted over this time span.

parameters to two, IgG response to N and IgM response to the spike protein (Fig 5B), was sufficient to identify COVID-19 samples, for all subjects except the aforementioned Subject i-0003. Furthermore, we had longitudinal data for a COVID-19 subject (Subject i-0004) who seroconverted by IgG over the course of eight days. Mapping the data from this subject shows that the longitudinal course of this subject's antibody response could clearly be mapped going from the pre-COVID-19 or non-infected region to the COVID-19 region.

Analysis of IgM and IgG seropositivity in COVID-19 subjects, with respect to time from first positive test and onset of symptoms (S2 Fig) showed that (1) seronegative results were only found in two cases (i-0003 and i-0004) where the sample was collected within two days of the onset of symptoms and (2) that all samples that were seropositive by IgM were also seropositive by IgG, as measured by response to the SARS-CoV-2 spike protein. This apparent simultaneous seroconversion was seen as early as two and three days after onset of symptoms (i-0009 and i-0005, respectively).

## Discussion

In the current study, we evaluated a new multiplex coronavirus antigen panel using an electro-chemiluminescence assay platform to conduct serological high-throughput testing of sera/plasma. The study had two objectives: (a) determine whether the methodology is useful for sero-surveillance and (b) to gain insights into serological cross-reactivity between five human coronaviruses. Analysis of the high-dimensional serological data (20 parameters per sample collected) revealed clear differences between pre-existing immunity and SARS-CoV-2 induced antibody responses and distinct patterns of cross-reactivity in IgM and IgG responses, demonstrating the value of this multiplex approach for SARS-CoV-2 serology studies.

The ECLIA-based MSD platform was chosen based on its superiority in previous studies using malarial antigens [16, 17]. In the present study, we evaluated a pan-CoV panel of recombinant proteins generated in a mammalian expression system to ensure proper glycosylation. We found that the linear range of the MSD ranged from 1:1,000 to 1:30,000, which eliminates the need to test serial dilutions for individual samples. An assay with such high sensitivity and specificity requires only very small sample volume (facilitating longitudinal studies), and is also more likely to detect SARS-CoV-2-specific antibodies for longer periods of time after

recovery. This is critical for serosurveillance approaches, particularly in light of recent studies which report sero-reversion within weeks to months of infection [18, 19]. Multiplexing the various antigens and testing only at one dilution provides significant sample sparing and increases the throughput of the assay. Another advantage of the MSD platform is the lack of apparent competition for antibody binding between related test antigens [16] due to their physical separation within the assay wells. Such competition has the potential to introduce significant artifacts when antigen-antibody binding occurs in a liquid phase as is the case in fluorescent bead-based flow cytometry (e.g. Luminex) [16]. This aspect is critical to the current study where the spike proteins of five CoVs are simultaneously being tested.

While the ECLIA assay tests reactivity to multiple antigens simultaneously, the assay must ultimately be validated against samples from individuals with known exposure to *each* antigen in the panel to determine thresholds for seropositivity and assess the specificity. Here we provide this validation for SARS-CoV-2 antigens using samples known to be exposed to SARS-CoV-2 and utilized a single threshold for defining seropositivity based on negative controls, but cross-reactivity in antibody responses between the CoV antigens necessitates individual validation of responses to each antigen to maximize specificity. This limitation is highlighted in the apparent 50% IgM seropositivity for MERS-CoV in the Control Group. While there was a MERS outbreak in South Korea in 2015, there were only 186 confirmed cases in that outbreak [20] and a more likely explanation is that this reflects a cross-reactivity from immunity to a related beta coronavirus. On a similar note, we were surprised to find 25% IgG seropositivity for SARS-CoV-2 in the Control group. Given that they were all seronegative to SARS-CoV-2 by IgM, this suggests the possibility of either a prior asymptomatic SARS-CoV-2 infection or cross-reactivity from immunity to another coronavirus.

Assessing the serological landscape of CoV-specific IgM and IgG responses resulted in several key observations: 1) IgG seropositivity to seasonal OC43 and HKU1 as well as influenza H3 was high, while IgM seropositivity to these antigens was low; 2) IgM seropositivity to SARS-CoV-2 was highly specific, with 90% seropositivity in COVID-19 samples and 0% seropositivity in Control or Pre-COVID samples; 3) SARS-CoV-2 IgG responses were highly cross-reactive with almost 60% of SARS-CoV-2 IgG seropositive samples being seropositive for *all* five CoV spike antigens; and 4) IgM and IgG SARS-CoV-2 spike responses appear to show different fine specificities, with IgM spike responses being largely recapitulated by the SARS-CoV-2 RBD antigen, while IgG spike responses were not. Taken together these observations suggest a few explanations. First, that the IgM response measured here largely reflect short-term antibody responses to acute or recent infections, while the IgG response here reflects long-term memory responses (in the case of the seasonal influenza H3, OC43, and HKU1) and/or later-stage, possibly affinity-matured, responses (in the case of COVID-19 samples). Accordingly, the early IgM response is highly specific to SARS-CoV-2 and focused on the RBD, while the late IgG response is broadly cross-reactive to many CoVs and includes non-RBD or RBD-adjacent epitopes. One immunological explanation for this pattern of responses is that the SARS-CoV-2 IgM response is naïve-derived and thus highly specific to SARS-CoV-2, while the IgG response is largely memory-derived, from cross-reactive B cells from prior CoV infections, and thus biased towards conserved or broadly cross-reactive SARS-CoV-2 epitopes.

Different origins of the CoV-2 IgM and IgG response (naïve vs. memory derived) could explain the apparent near-simultaneous emergence of IgM and IgG responses [21], lacking the interval period thought to be associated isotype-class switching in a primary infection, that has also been observed in SARS in 2003 [22]. Wec et al [23] showed that the memory B cell repertoire from an individual that survived SARS-CoV-1 infection in 2003 contained hundreds of B cells that were broadly neutralizing across multiple human CoVs, including CoV-2, suggesting

they derived from memory B cells to prior CoV infections. Further corroborating evidence is found by Ng et al. [24] who found that approximately 10–20% of pre-pandemic or non-CoV-2 infected samples showed CoV-2-reactive IgG responses while none showed CoV-2-specific IgM responses, very similar to our findings. They found that these cross-reactive CoV-2-reactive IgG antibodies largely target the more highly conserved spike S2 domain, not the S1 domain that contains the RBD, while CoV-2 infection induced IgG and IgM antibodies target both S1 and S2 domains, supporting the theory that pre-existing CoV immunity is largely biased towards conserved S2 epitopes. Finally, our findings suggest that for antibody-based diagnostics and serosurveillance, IgM and CoV-2-N-specific responses may have higher specificity than IgG and CoV-2-spike-responses, and that RBD-specific IgG responses in particular might have poor sensitivity in individuals with COVID-19 whose IgG response is largely derived from pre-existing CoV immunity focused on conserved S2 epitopes.

In this study, we demonstrate that by combining IgM and IgG responses to spike and N proteins, the ECLIA assay platform is able to reliably distinguish COVID-19 samples from Control or Pre-COVID-19 samples. IgM responses alone were found to be highly specific, but may have limited durability, while IgG responses were less specific, but potentially more long-lived–possibly distinguishing acute infection from convalescence or prior exposure. Furthermore, IgG responses of COVID-19 patients were more cross-reactive with spike proteins of other CoVs. This is an important finding since most reports on SARS-CoV-2 serology focus on assessing the level of SARS-CoV-2 specific IgG. Our findings support the strategy of some point of care antibody testing kits that assess IgM and IgG to identify ongoing/recent infection or previous exposure [7]. The fact that our unsupervised approach to combining IgM and IgG responses was able to distinguish COVID-19 subjects suggests that a machine-learning approach using a larger data set would have high potential for detecting acute infection status and prior exposure of an individual from their serological data.

There were several limitations to the present study. First, the sample size is relatively small and as such the study is intended primarily to demonstrate feasibility of the multiplex ECLIA assay. Second, the samples were obtained through a public health investigation of a local outbreak in Camp Humphreys, and thus largely consists of 'convenience' samples. While we matched Control subjects to the same location and study population, a rigorous case-control study was infeasible in the midst of an emergency outbreak response. Still, the samples reflect diversity in disease onset and severity that parallels samples collected in real-world serosurveillance efforts. Third, with some exceptions, the study did not include longitudinal sample collection which limits its findings with respect to disease progression. Fourth, while COVID-19 and Control groups were matched by site and population, pre-pandemic samples were obtained from a sample collection protocol carried out domestically, in Maryland, and thus provides an imperfect pre-pandemic comparison to the pandemic samples.

In summary, the new multiplex assay demonstrated the power of assessing both, IgM and IgG specific for pan-CoVs—and SARS-CoV-2 in particular—and showed the power of this readout to establish serological landscapes that contribute to our understanding of the role of cross-reactivities between the various CoV and the impact on immunity and protection. Furthermore, the present study also demonstrates the power of the MSD multiplex platform in quickly establishing serological profiles of specific populations and cohorts to guide vaccine design and optimization and identify biomarkers of immunity or disease.

## Supporting information

**S1 Fig. Sensitivity and specificity of assay to detect SARS-CoV-2-specific antibodies.** (PDF)

**S2 Fig. IgM and IgG seropositivity with respect to disease progression.**
(PDF)

**S1 Table. Age and sex of control subjects.**
(PDF)

**S2 Table. Serological dataset.**
(XLSX)

## Acknowledgments

The authors would like to thank Ms. Elizabeth Duncan for technical assistance at various levels of the project. This study could not have been completed without the help of an excellent team of laboratory support personnel and other project staff working overseas who helped obtain and transport samples. Laboratory and project staff include: MAJ Ashley Torrence, SPC Bryanna Harris, SPC Christian Stevens, SPC Jacob Lacourse, SSG Jessie Rodriguez, CAPT Robert Pilla, CPT Scott Kim, SGT Taylor Wolik Finally, we express our sincere appreciation to all those volunteers that provided samples for this project.

## Declarations

ESB-L is a government employee. Title 17 U.S.C. § 105 provides that "Copyright protection under this title is not available for any work of the United States Government, but the United States Government". Title 17 U.S.C. § 101 defines US Government work as "work prepared by a military service member or employee of the US Government as part of that person's official duties".

## Disclaimer

Material has been reviewed by the Walter Reed Army Institute of Research. There is no objection to its presentation and/or publication. The opinions or assertions contained herein are the private views of the authors, and are not to be construed as official, or as reflecting the views of the Department of the Army or the Department of Defense. The investigators have adhered to the policies for protection of human subjects as prescribed in AR 70–25. This paper has been approved for public release with unlimited distribution.

## Author Contributions

**Conceptualization:** Elke S. Bergmann-Leitner.

**Formal analysis:** Sidhartha Chaudhury, Jessica S. Bolton, Elke S. Bergmann-Leitner.

**Funding acquisition:** Jason A. Regules, Clinton K. Murray, Kayvon Modjarrad.

**Investigation:** Evelyn Mose, William O'Connell, Joseph Hudak, Paul Scott.

**Methodology:** Jessica S. Bolton, Elke S. Bergmann-Leitner.

**Project administration:** Jack Hutter, Shilpa Hakre, Amy Wooten, Jason A. Regules, Clinton K. Murray.

**Resources:** Jack Hutter, Shilpa Hakre, Evelyn Mose, Amy Wooten, William O'Connell, Joseph Hudak, Janice M. Darden, Kayvon Modjarrad, Paul Scott.

**Software:** Sidhartha Chaudhury.

**Supervision:** Jason A. Regules, Clinton K. Murray, Elke S. Bergmann-Leitner.

**Writing – original draft:** Sidhartha Chaudhury, Elke S. Bergmann-Leitner.

**Writing – review & editing:** Sidhartha Chaudhury, Jack Hutter, Jessica S. Bolton, Shelly J. Krebs, Jason A. Regules, Clinton K. Murray, Paul Scott, Sheila Peel.

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
