## [Decision Letter · Decision Letter 0]

3 Feb 2021

PONE-D-21-00516

Serological Profiles of Pan-CoV Responses in COVID-19 Patients Using a Multiplexed Electro-chemiluminescence-based Testing Platform

PLOS ONE

Dear Dr. Bergmann-Leitner,

Thank you for submitting your manuscript to PLOS ONE. After careful consideration, we feel that it has merit but does not fully meet PLOS ONE’s publication criteria as it currently stands. Therefore, we invite you to submit a revised version of the manuscript that addresses the points raised during the review process.

Both reviewer find the article of interest, even if many improvements are suggested by the reviewer 1, the results remained important even if a full description of the study limitations deserved to be added.

The main concern of the two reviewers, however, is the choice of the naïve population from a very different epidemiologic context: US population versus a Korean military group). This deserved to be adressed carefully and I suggest to add some data from a pre-epidemic Korean population or even from people previously exposed to other corona viruses as it is expected to be found in Korea.

We look forward to receiving your revised manuscript.

Kind regards,

Pierre Roques, Ph.D.

Academic Editor

PLOS ONE

Journal Requirements:

2.) Please provide a sample size and power calculation in the Methods, or discuss the reasons for not performing one before study initiation.

3.) Supplementary materials are referenced in your manuscript but appear to be missing. Please upload these as supplementary files.

4.) PLOS ONE requires experimental methods to be described in enough detail to allow suitably skilled investigators to fully replicate and evaluate your study.

See https://journals.plos.org/plosone/s/submission-guidelines#loc-materials-and-methods for more information.

To comply with PLOS ONE submission guidelines, in your Methods section, please provide:

a) the sources, catalog numbers, and dilutions of the SULFO-TAG antibodies used in your study

b) the catalog/identifying numbers for the two commercially available pooled plasma samples

c)  the sequences or accession numbers of the antigens used in your study.

5.) Thank you for stating the following in the Funding Section of your manuscript:

'This work was supported by the Military Infectious Disease Research Program (MIDRP).'

'The funders had no role in study design, data collection and analysis, decision to publish, or preparation of the manuscript.'

6.) We note that you have indicated that data from this study are available upon request. PLOS only allows data to be available upon request if there are legal or ethical restrictions on sharing data publicly. For more information on unacceptable data access restrictions, please see http://journals.plos.org/plosone/s/data-availability#loc-unacceptable-data-access-restrictions.

7.) Your ethics statement should only appear in the Methods section of your manuscript. If your ethics statement is written in any section besides the Methods, please move it to the Methods section and delete it from any other section. Please ensure that your ethics statement is included in your manuscript, as the ethics statement entered into the online submission form will not be published alongside your manuscript.

8.) Please include captions for your Supporting Information files at the end of your manuscript, and update any in-text citations to match accordingly. Please see our Supporting Information guidelines for more information: http://journals.plos.org/plosone/s/supporting-information

9.) Please amend either the title on the online submission form (via Edit Submission) or the title in the manuscript so that they are identical.

Reviewers' comments:

Reviewer's Responses to Questions

**Comments to the Author**

1. Is the manuscript technically sound, and do the data support the conclusions?

Reviewer #1: No

Reviewer #2: Yes

2. Has the statistical analysis been performed appropriately and rigorously? 

Reviewer #1: I Don't Know

Reviewer #2: Yes

3. Have the authors made all data underlying the findings in their manuscript fully available?

Reviewer #1: Yes

Reviewer #2: Yes

4. Is the manuscript presented in an intelligible fashion and written in standard English?

Reviewer #1: Yes

Reviewer #2: Yes

5. Review Comments to the Author

Reviewer #1: *Summary of the research

The clinical presentation of covid-19 (+) is wide, inconstant ad emerging, making the diagnostic, management and any implementation difficult. In response to that, the manuscript claims to value a new 10-plex electro-chemiluminescence assay as a multitasking Covid-19 screening platform. The dual IgM/IgG detection, respectively targeting Covid-19 Spike Covid-19 NP proteins, allows to achieved high specificity and sensitivity levels with a low samples-consuming. The cross reactivity is also considered in diagnosis to distinguish the covid-19 (+) persons (ill or cured) from the healthy population and/or covid-19 (-) persons and be a high-throughput sero-surveillance tool of quality.

*Overview of the manuscript’s

The multiplex approach is a commonly tool used in serological survey, including SARS-Cov2 as here : 1) The IgG/IgM detection allows to determine host responses profile independently to clinical signs and compared to control group, 2) The SARS-Cov-2 10-plex raises on the cross reactivity trouble. Nevertheless, this lacks of creativity " with a feeling of "déjà vu". The choice of a ECLIA approach remains unclear in implementation: diagnostic (combined with RT-qPR), management or survey (prevalence)?

Several papers already demonstrated the multiplex tools accuracy (%specificity and % sensitivity) to IgM and/or IgG detection in various population groups sera. Nevertheless, at the end of the discussion, the goal seems not to be fully achieved. Despite adequate statistical analysis, the data seems not enough exploited to draw the supported conclusions.

As reporting in others similar studies, the cross-reactivity concern between coronavirus is also discussed here. Its impact in the tools performance and the developing some plateforms depends on targeted population by covid-19. The sampling selection process and the choice of population categories are not enough justified. The groups detailed description lack and reducing the interpreted accurate. The authors forgot to address some possible limitations of the research such as a military environment choice: healthy and male person in majority.

To sum up, the improvement involves in giving details of the sampling collection process and providing more patients information. Analyses and discussion need to enrich in order to link the findings of this study to the supported conclusions (confer below recommendations).

*Recommended course of action (major)

1-Population study and selection criteria, and sampling:

-The choice of population groups and selection criteria support the study quality. The covid-19 and non-COVID-19 patients are enlisted at the same site and in the pandemic time period. However, the sampling process needs to be precise. The selection criteria has to be the same between these both populations in order to perform comparative study.

-Moreover, the negative status of non-Covid-19 patient (working in health center) seems only based on the observation (clincal forms) (evaluation bias) and/or the paitent declaration during the interview (no risk of exposition?) (= memory bias). While the prevalence of COVID-19 in Health Care Worker is low, providing evidence of (RT-qPCR) tests for non-covid-19 group is recommended.

-The military (covid-19 group) as well as the health care persons (non covid-19 group) are often considering as healthy and volunteer persons in participating in study compared to the general population. This behavior may impact on serological results and needs to be taking into consideration (analyses and discussion).

-The selection of pre-pandemic samples from several sources in the United States in 2019 is also questioned (Random or voluntary strategies?). In addition, epidemic information about coronavirus circulation during this 2019 summer would allow to determine whether the exposition to others CoV species differs between pre-pandemic samples and pandemic samples (Covid-19 (+) and Non-Covid-19 (-)) and enrich the cross-reactivity analyses.

-The lack of homogeneity in the inclusion criteria between 3 groups making difficult the value of antibody responses and their interpretation.

2-Sex and ages:

As influencing the antibodies response (i.e.= kinetic, duration, level), the sex and ages need to be noticed in groups.

3-Results graphic:

In addition to table 1, the illustration of patient features through a timeline-scale would better link the diseases progression, onset with the antibody responses and potentially other determinants factors.

*Recommended course of action (minor)

1-Severity level determination:

The illness definition and the levels severity remain uncleared for the covid-19 group: is it related to a score scale and performed by a same medical person (table 1: page 186)?

2-Chapter reorganisation:

The 334-347 discussion paraph, including particular objectives, should be moved in the introduction than in the conclusion to facilitate the results analyses and authors goals.

-Results presentation in only 2 major parts "IgM/IgG characterization" and "the cross reactivity events" might facilitate and open the discussion and subchapters could be suggested.

*Specific areas for improvement in discussion

1-Onset diseases importance or delay in clinical signs apparition:

-The ELCIA approach allows to characterize the COVID-19 samples compared to pre-pandemic and control based on IgM/IgG detection with a high sensibility (100- to 200-fold higher).

The authors claim that by using a larger data set, ELCIA would have high potential for predicting acute infection status and exposure of an individual from their serological data. If is it true: how to explain the difference between id-003 and id-009? Indeed, the id-0003 and i-0009 subjects showed the same mild clinical presentation and the timeline of samples collection.

As arguing the authors, the absence of antibodies response to Subject i-0003 may be related to mild clinical presentation mild symptoms and/or the delay of serconversion (only one day following initial test positivity). If so, what about the interpretation of the id-009 results, showing similar clinical profile as the id-0003 patient (215-218)?

-As mentioned above, the authors suggests that the delay in antibodies response is related to the low clinical symptom: How explaining the IgM and IgG profiles of the id-007, id-008, id-0010? = asymptomatic patients whose samples, collected + 2 days after PCR (+), are associated to significant antibodies response?

-To argue in the IgM/IgG conjugated detection importance in Covid-19 management and/or survey, discussion should be enriched with additional related references.

2-Cross-reactivity:

The authors address possible limitations of the research, including cross-reactivity

To gain insights into the pre-existing immunity Covid induced antibody responses impact, the distinct patterns of cross-reactivity in IgM and IgG responses are plenty explored in the literature, such as the multiplex tools use, as here. The cross reactivity appears higher with IgG SARS-CoV-2 than the IgM SARS-CoV-2 binding antibodies. The author provided a long analysis in the cross reactivity trouble between coronavirus species. SARS-CoV-2 IgG antibodies may be cross-reactive with SARS-CoV-1 and MERS-CoV: 1) 33% of CoV-2-seropositive COVID-19 samples were also seropositive for MERS-CoV, 2) 50% IgM seropositivity for MERS-CoV was detected the Control Group (275-277).

In view of the % in MERS/SARS COV-2 cross reactivity, the MERS serological status of control population should be clarify : Is it a lack in specificity of the ECLIA approach or a real high MERS prevalence in control populations?

3-Confunding factors?:

As mentioning in the results revision, details about patients (sex and age) may help authors as well as reviewers by arguing some exceptions instead of criticizing the tools accuracy.

The cross relationship between antibodies profile and clinical presentation needs to be discussed according to some host determinants. The sampling design impacts also in the way.

-Example concerning sex determinant: the proportion of female is lower in the military population compared to general population

-Example concerning military group: this population is better physical fitness, influencing the host responses to pathogen.

-Example concerning control group: even not being directly in contact to Covid (+) patients, people working in hospital have usually better style of life than the general population (or take better care of their health).

All of these justify the above revision request by giving details about the study of population and arguing on the criteria selection etc...

*To go to further…

By combing IgG and IgM results, the authors success in mapping the course of this subject’s antibody response from the pre-COVID-19 or non-infected region to the COVID-19 region independently to clinical symptoms. This view may help in vaccine implementation as well as vaccine efficacy. Nevertheless, this is not fully achieved in view of the id-003 antibody response (low antibody response, low duration or delay?). The results of serological profile of id-0003 need to be checked with others tools (ELISA commercial kit or ELISA in house) to conclude on the ELCIA tool performance/limitation.

The id-003 data calls to future research on IgA profile and/or seroneutalization. It should be important to mention that the diseases progression depends on the seroneutralization antibodies pool as well as the IgA kinetics (i.e. IgA secretion in local).

Unfortunately, the authors don’t open up enough perspectives.

*Current cutting-edge tool: is it? and for what?

The authors could bring more explanations in why: 1) this new multiplex coronavirus antigen evaluation may change the covid-19 survey; 2) the electro-chemiluminescence assay platform (serological high-throughput testing of sera/plasma) may support the current molecular covid-19 diagnosis?

*Additional minor points:

1-Rewrite the abstract

Shorter by highlighting the goal and the achievement or not of the study

2-Additional references

In introduction, the authors explain in why the study matters and put the research in context: antibodies response (kinetics of apparition, the delay in apparition, the lack of duration).

But they don’t precise in why ECLIA approach may be revolutionary compared to others?

Literature lacking in other serological approaches (ELISA gold standard, etc...) as well as about the comparative studies between various serological tools (ELISA vs ECLIA).

*Keys list of the paper

-Strengths:

Ethical guidelines respect

Exhaustive protocol description

Adequate statistical analyses

Graphics support the findings

-Weaknesses:

No creative concept or idea, “déjà vu concept”

Absence in population samples information and unclear election criteria

Scant or incomplete data explanation to draw conclusion

Non results synthesize prior discussion

Reviewer #2: Reviewing report for PLoS One PONE-D-21-00516 : “Serological Profiles of Pan-CoV Responses in COVID-19 Patients Using a Multiplexed Electro-chemiluminescence-based Testing Platform” from S. Chaudhury et alii.

In this report, S Chaudhury and coworkers exposed the outcome of a multiplexed electro-chemiluminescence-based assay (ECLIA) detecting pan-coronavirus immune response in the context of the COVID-19 pandemic. The assay has the capacity to detect IgM ad IgG response against full-length spike of various coronavirus and epitope specificity of the response against spike protein subdomains including receptor bind domain (RBD).

The study was conducted with COVID-19 patients and controls recruited in Korea while a third group of pre-pandemics patients was recruited in the US. Each group was composed of 10 patients. These rather small size is compensated by the depth of data analysis performed by the authors.

An important outcome of this study was that authors did not observed any association between seropositivity and disease severity.

The authors observed that in favorable circumstances, the ECLIA test is very sensitive allowing a dilution of up to 30.000. This high sensitivity was however incapable to detect a specific immune response in a COVID-19 patients, sampled 24H after initial diagnosis. This patient was virtually useless in the whole study (except to remind that serology should not be performed too early post disease onset).

The ECLIA revealed a substantial cross-reactivity of IgG between SARS-CoV-2 and various other coronaviruses either epidemic (SARS-CoV-1, MERS-CoV) or endemic (HKU1 and OC43). The differences between IgM and IgG specificity explain why the authors suggest that a machine-learning approach using a large set of data should be used to correctly diagnose a recent or acute exposure to SARS-CoV-2 (line 424). This complexity suggests that the interpretation of ECLIA is plausibly possible only for a given geographical context (ie exposed to a broadly common infectious environment) using controls coming from the same place than putative patients to be diagnosed. This aspect is not really developed by the authors. I suggest that they add a comment in the final version about the origin of these pre-pandemic plasmas as a potential limitation of their work (South Korean pre-COVID might have yielded an IgG cloud on figure 3 much closer or even overlapping with the COVID-19 cloud).

Overall, in the context of COVID-19 pandemics, the work is interesting and bring valuable insight about the various components building the immune response against SARS-CoV-2. It emphasizes as well the technical versatility of ECLIA (linear range, scalability, etc…).

The reader might regret that pre-pandemic samples have been collected from subjects living in the US, a country that, at variance with South Korea, was not as affected by previous SARS-CoV-1 and MERS-CoV epidemics. However this situation does not seem to alter the overall conclusions of the paper despite the absence of any long-term seroreactivity to MERS-CoV or SARS-CoV-1 in pre-pandemic plasmas

Minor issue;

Line 216: “his plasma samples” not “their”

Line 364: “ECLIA”, “must be validated”

Figure 3: IgG seropositivity, the smallest-greenish piece of the pie represents 10% not 20%.

6. PLOS authors have the option to publish the peer review history of their article (what does this mean?). If published, this will include your full peer review and any attached files.

Reviewer #1: No

Reviewer #2: **Yes: **Pascal Pineau

---

## [Author Response · Author response to Decision Letter 0]

11 Feb 2021

Detailed responses to the editorial team:

• Funding statement:

The work was funded by the Military Infectious Disease Research program (MIDRP), which was not in the online database of funders. The funders did not have any influence on this study and the experimental plan.

• Revision of figures

We have reformatted the figures using the PACE website and uploaded the revised files.

• Details of experimental work

We have updated our protocol to include catalog numbers and other details as outlined below. It should be noted that we have followed the manufacturer’s (Mesoscale) instructions for the qualified assay. We now describe the computational analysis in the manuscript. If the reviewers/editor feel that the readers would benefit from either portion of the experimental work or the computational analysis being further detailed in a separate protocol, then we will deposit a description in protocols.io.

(1) Please make sure that your manuscript meets PLOS ONE’s style requirements.

We have made the requested corrections.

(2) Please provide a sample size and power calculation in the Methods, or discuss the reasons for not performing one before study initiation.

We have added the following statement to the Materials Section (lines 129-133): “Samples from 10 COVID-19 subjects and 8 Control subjects, matched by study location and population, were obtained and compared to a similar number of pre-pandemic samples (10) using the multiplex ECLIA assay. As this was a retrospective analysis of COVID-19 samples collected during a public health investigation of a local outbreak, no a priori power calculation was carried out."

(3) Supplementary materials are referenced in your manuscript but appear to be missing. Please upload these as supplementary files.

We apologize for the omission – an error during the initial submission process. We have uploaded the missing figure S1 Fig, added new supplementary materials; (1) a supplementary S1 Table (demographics of the population control as well as pre-COVID-19 samples), and (2) S2 Fig (IgM and IgG seropositivity with respect to disease progression); (3) spreadsheet with the raw serological data for readers to download and reproduce our results.

(4) To comply with PLOS ONE submission guidelines, in your Method section, please provide

a) the source, catalog numbers, and the dilution of the SULFO-TAG antibodies in your study

line 169 – we have added the details on the antibodies: “…SULFO-TAG either with anti-human IgG (Cat.No D20JL, MSD) antibody or anti-human IgM (Cat.No D20JP, MSD) was diluted to 2 μg/ml in Diluent 100…”

b) the catalog/identifying numbers for the two commercially available pooled plasma samples:

line 143 – we have added the order information (GeminiBio (GemCell™ U.S. Origin Human Serum AB, Cat.No 100-512). We obtained the items to test them for their ability to support tissue cultures and received several lots to test in 2018. These two samples were aliquots of this lots.

c) Sequence or accession numbers of the antigens used in your study.

We have added the requested information (lines 154-159).

5) Funding information

As mentioned above, the work was funded by MIDRP and we have deleted the section in the acknowledgements. We would appreciate if we could revise this in the online submission system.

6) Data access

We have revised the statement to:” The serological data (expressed as Luminescence signal) are provided in S2 Table. These data are raw data and were log-transformed for the downstream analysis described in the manuscript” Lines 478-479

7) Ethics statement

We have removed the section and added the information to the Materials section (lines 145-149).

8) Please include captions for your Supporting Information files at the end of your manuscript, and update any in-text citations to match accordingly.

We have made the requested revisions.

Discrepancy in titles between manuscript and submission system

We have adjusted the title in the submission system to reflect the title in the manuscript file.

REVIEWER 1 COMMENTS: Major Points:

1) Population study and selection criteria, and sampling: The covid-19 and non-COVID-19 patients are enlisted at the same site and in the pandemic time period. However, the sampling process needs to be precise. The selection criteria has to be the same between these both populations in order to perform comparative study.

Control and COVID-19 subjects were recruited from the same site and study population (U.S. Department of Defense military, civilian, and contractor population working at Camp Humphreys). We updated the Methods to reflect this:

“COVID-19 and Control subjects were drawn from the same overall population: the U.S. Department of Defense military, civilian, and contractor population working at Camp Humphreys.” Lines 124 ff

We also added a paragraph in the Discussion section highlighting the limitations to the present study (lines 446 ff):

“There were several limitations to the present study. First, the sample size is relatively small and as such the study is intended primarily to demonstrate feasibility of the multiplex ECLIA assay. Second, the samples were obtained through a public health investigation of a local outbreak in Camp Humphreys, and, thus, largely consists of ‘convenience’ samples. While we matched Control subjects to the same location and study population, a rigorous case-control study was infeasible in the midst of an emergency outbreak response. Still, the samples reflect diversity in disease onset and severity that parallel samples collected in real-world serosurveillance efforts. Third, with some exceptions, the study did not include longitudinal sample collection, which limits its findings with respect to disease progression. Fourth, while COVID-19 and Control groups were matched by site and population, pre-pandemic samples were obtained from a sample collection protocol carried out domestically, in Maryland, and ,thus, provides an imperfect pre-pandemic comparison to the pandemic samples.”

2) The negative status of non-Covid-19 patient (working in health center) seems only based on the observation (clincal forms) (evaluation bias) and/or the paitent declaration during the interview (no risk of exposition?) (= memory bias). While the prevalence of COVID-19 in Health Care Worker is low, providing evidence of (RT-qPCR) tests for non-covid-19 group is recommended.

Control samples were tested for COVID-19 using the same diagnostic assay as the COVID-19 patients. We regret this omission, and it is now included in the Methods (Line 134-135):

“All Control subjects were also tested via nasopharyngeal swab and RT-PCR and confirmed to be negative for COVID-19 at the time of sample collection.”

3) The military (covid-19 group) as well as the health care persons (non covid-19 group) are often considering as healthy and volunteer persons in participating in study compared to

the general population. This behavior may impact on serological results and needs to be taking into consideration (analyses and discussion).

We have included age range and sex for COVID-19 and Control subjects in Table 1 and in S1 Table. Subjects in this study were not exclusively military, and included U.S. Department of Defense civilians and contractors thus reflecting a wide demographic range.

4) The selection of pre-pandemic samples from several sources in the United States in 2019 is also questioned (Random or voluntary strategies?). In addition, epidemic information about coronavirus circulation during this 2019 summer would allow to determine whether the exposure to others CoV species differs between pre-pandemic samples and pandemic samples (Covid-19 (+) and Non-Covid-19 (-)) and enrich the cross-reactivity analyses.

Selection of pre-pandemic samples was based purely on availability of samples obtained from a prior WRAIR blood collection protocol, and was thus effectively random, within the geographic and demographic constraints of the local population. We updated the method to reflect that:

“Pre-pandemic samples were obtained from a WRAIR blood collection protocol (WRAIR#2567), based on sample availability, from August 2019 conducted in Silver Spring, Maryland.” Lines 140 ff

Since the COVID-19 and Control samples were collected from a Department of Defense military and civilian population that is relatively highly mobile, it would not be possible to determine general trends in pre-pandemic coronavirus exposure in these individuals with diverse geographic history.

5) The lack of homogeneity in the inclusion criteria between 3 groups making difficult the value of antibody responses and their interpretation. Sex and ages: as influencing the antibodies response (i.e.=kinetic, duration, level), the sex and ages need to be noticed in the group.

Sex and age range in COVID-19 group and Control group now included in Table 1 and S1 Table respectively.

6) In addition to table 1, the illustration of patient features through a timeline-scale would better link the diseases progression, onset with the antibody responses and potentially other determinants factors.

We have now added a timeline for the COVID-19 subjects that includes symptomology, testing, and seropositivity as determined by the ECLIA assay in S2 Fig. We added the following paragraph to the Results section (lines 342 ff):

“Analysis of IgM and IgG seropositivity in COVID-19 subjects, with respect to time from first positive test and onset of symptoms (S2 Fig) showed that (1) seronegative results were only found in two cases (i-0003 and i-0004) where the sample was collected within two days of the onset of symptoms and that (2) all samples that were seropositive by IgM were also seropositive by IgG, as measured by response to the SARS-CoV-2 spike protein. This apparent simultaneous seroconversion was seen as early as two and three days after onset of symptoms (i-0009 and i-0005, respectively).”

Minor points:

• Severity level determination: The illness definition and the levels severity remain uncleared for the covid-19 group: is it related to a score scale and performed by a same medical person (table 1: page 186)?

We provided additional detail in the Methods section to address this question:

“COVID-19 disease severity was assessed as asymptomatic, mild (symptomatic, but not interfering with daily activity), moderate (interfering with daily activity, but not requiring hospitalization), and severe (preventing daily activity and requiring hospitalization).”

• The 334-347 discussion paraph, including particular objectives, should be moved in the introduction than in the conclusion to facilitate the results analyses and authors goals.

We removed that paragraph as we felt it was redundant with material in the Introduction

• Results presentation in only 2 major parts "IgM/IgG characterization" and "the cross reactivity events" might facilitate and open the discussion and subchapters could be suggested.

We opted to keep the Results section organization as is since sub-sectioning the results would fragment the data presentation too much.

• The authors claim that by using a larger data set, ELCIA would have high potential for predicting acute infection status and exposure of an individual from their serological data. If is it true: how to explain the difference between id-003 and id-009?

The assay was able to detect seroconversion in 3 of 5 cases where samples were collected within seven days of symptom onset (S2 Fig, bottom panel), showing that detecting acute infection using this assay is possible, even in mild cases. The observation that not all samples collected in this time span showed seroconversion highlights the limitations of using a serology assay to detect early infection.

• As arguing the authors, the absence of antibodies response to Subject i-0003 may be related to mild clinical presentation mild symptoms and/or the delay of serconversion (only one day following initial test positivity). If so, what about the interpretation of the id-009 results, showing similar clinical profile as the id-0003 patient (215-218)?

It is common for different individuals to have different time courses with respect to disease progression and seroconversion. Therefore, it is not unusual for one subject to seroconvert one day before another subject with a similar clinical profile.

• As mentioned above, the authors suggests that the delay in antibodies response is related to the low clinical symptom: How explaining the IgM and IgG profiles of the id-007, id-008, id-0010? = asymptomatic patients whose samples, collected + 2 days after PCR (+), are associated to significant antibodies response?

It is difficult to assess disease progression in asymptomatic subjects because they do not have a time of onset of symptoms from which to compare with other subjects. Time of first positive test in asymptomatic subjects is, to some extent, arbitrary, based on when external circumstances (routine testing, contract tracing, etc.) prompted them to get tested.

• To argue in the IgM/IgG conjugated detection importance in Covid-19 management and/or survey, discussion should be enriched with additional related references.

We believe that integrating multiple serological measurements in serosurveillance and diagnostic efforts is quite novel, very powerful, and not yet established in the literature, and we hope that this study will spur further research efforts in this direction.

• The authors address possible limitations of the research, including cross-reactivity To gain insights into the pre-existing immunity Covid induced antibody responses impact, the distinct patterns of cross-reactivity in IgM and IgG responses are plenty explored in the literature, such as the multiplex tools use, as here.

We could not find other examples of published studies exploring cross-reactivity of IgM and IgG responses in COVID-19, except those already cited in the Discussion section.

• The cross reactivity appears higher with IgG SARS-CoV-2 than the IgM SARS-CoV-2 binding antibodies. The author provided a long analysis in the cross reactivity trouble between coronavirus species. SARS-CoV-2 IgG antibodies may be cross-reactive with SARS-CoV-1 and MERS-CoV: 1) 33% of CoV-2-seropositive COVID-19 samples were also seropositive for MERS-CoV, 2) 50% IgM seropositivity for MERS-CoV was detected the Control Group (275-277). In view of the % in MERS/SARS COV-2 cross reactivity, the MERS serological status of control population should be clarify : Is it a lack in specificity of the ECLIA approach or a real high MERS prevalence in control populations?

We stated in the Discussion that we believed it was the lack of specificity in the ECLIA and that validation with each CoV antigen was necessary to establish threshold values for seropositivity (lines 384 ff):

“Here we provide this validation for SARS-CoV-2 antigens using samples known to be exposed to SARS-CoV-2 and utilized a single threshold for defining seropositivity based on negative controls, but cross-reactivity in antibody responses between the CoV antigens necessitates individual validation of responses to each antigen to maximize specificity. This limitation is highlighted in the apparent 50% IgM seropositivity for MERS-CoV in the Control Group. While there was a MERS outbreak in South Korea in 2015, there were only 186 confirmed cases in that outbreak [22] and a more likely explanation is that this reflects a cross-reactivity from immunity to a related beta coronavirus.”

• Confunding factors: As mentioning in the results revision, details about patients (sex and age) may help authors as well as reviewers by arguing some exceptions instead of criticizing the tools accuracy.

We analyzed for significant differences in age-range and sex and did not find any. A study of this size is not sufficiently powered to identify significant differences in serological data with respect to sex and age.

• The cross relationship between antibodies profile and clinical presentation needs to be discussed according to some host determinants. The sampling design impacts also in the way. -Example concerning sex determinant: the proportion of female is lower in the military population compared to general population -Example concerning military group: this population is better physical fitness, influencing the host responses to pathogen. -Example concerning control group: even not being directly in contact to Covid (+) patients, people working in hospital have usually better style of life than the general population (or take better care of their health). All of these justify the above revision request by giving details about the study of population and arguing on the criteria selection etc...

The COVID-19 and Control subject population in this study are not exclusively military and includes civilians and contractors and age and sex are now provided in Table 1 and S1 Table. A systematic assessment of differences in serology between military and civilian populations is outside of the scope of this effort. Regarding the concern related to the lifestyle of the study population not being representative of that of the general population it should be noted that they represent individuals working at a military base in various functions, and are not necessarily healthcare workers).

• By combing IgG and IgM results, the authors success in mapping the course of this subject’s antibody response from the pre-COVID-19 or non-infected region to the COVID-19 region independently to clinical symptoms. This view may help in vaccine implementation as well as vaccine efficacy. Nevertheless, this is not fully achieved in view of the id-003 antibody response (low antibody response, low duration or delay?). The results of serological profile of id-0003 need to be checked with others tools (ELISA commercial kit or ELISA in house) to conclude on the ELCIA tool performance/limitation.

The reviewer points out an important aspect of such analyses: It is not uncommon to find COVID-19 subjects that have not seroconverted, particularly early in the disease course. The sample for subject i-0003 was collected one day after symptom onset. The ECLIA assay is much more sensitive than a regular ELISA (we had previously published formal, extensive comparisons of the platforms) and in the absence of an ECLIA signal, it is highly unlikely that any ELISA would be able to detect a signal.

• The id-003 data calls to future research on IgA profile and/or seroneutalization. It should be important to mention that the diseases progression depends on the seroneutralization antibodies pool as well as the IgA kinetics (i.e. IgA secretion in local). Unfortunately, the authors don’t open up enough perspectives.

We agree with the reviewer that these functional readouts will provide additional, important insights, but would like to point out that binding-based antibody assays are common in both

diagnostic and serosurveillance studies. The relationship between binding and neutralization is a substantial undertaking that is well outside the scope of the present study. It would be appropriate for a follow-up study that will require a different sample collection protocol.

• The authors could bring more explanations in why: 1) this new multiplex coronavirus antigen evaluation may change the covid-19 survey; 2) the electro-chemiluminescence assay platform (serological high-throughput testing of sera/plasma) may support the current molecular covid-19 diagnosis?

We describe in the Discussion that a multiplex assay has the potential for greater specificity in serosurveillance over standard single-antigen approaches and also enables for characterization of immunity across multiple coronaviruses to determine if and how such immunity might contribute to COVID-19. In addition, the approach we describe enables significant sample sparing and a high level of reproducibility.

REVIEWER 2 COMMENTS:

(No major comments, minor comments addressed in manuscript text

---

## [Decision Letter · Decision Letter 1]

10 May 2021

PONE-D-21-00516R1

Serological profiles of pan-coronavirus-specific responses in COVID-19 patients using a multiplexed electro-chemiluminescence-based testing platform

PLOS ONE

Dear Dr. Bergmann-Leitner,

Thank you for submitting your manuscript to PLOS ONE. After careful consideration, we feel that it has merit but does not fully meet PLOS ONE’s publication criteria as it currently stands. Therefore, we invite you to submit a revised version of the manuscript that addresses the points raised during the review process.

The two independent reviewers raised the completely opposite comments: accept and reject. Thus, I asked the third reviewer to give us the new comments, and it was accept. So please make a rebuttal against the comments provided from the reviewer 1. I will give a decision for the publication.

We look forward to receiving your revised manuscript.

Kind regards,

Etsuro Ito

Academic Editor

PLOS ONE

Journal Requirements:

Reviewers' comments:

Reviewer's Responses to Questions

**Comments to the Author**

1. If the authors have adequately addressed your comments raised in a previous round of review and you feel that this manuscript is now acceptable for publication, you may indicate that here to bypass the “Comments to the Author” section, enter your conflict of interest statement in the “Confidential to Editor” section, and submit your "Accept" recommendation.

Reviewer #1: (No Response)

Reviewer #2: All comments have been addressed

Reviewer #3: All comments have been addressed

2. Is the manuscript technically sound, and do the data support the conclusions?

Reviewer #1: Partly

Reviewer #2: Yes

Reviewer #3: Yes

3. Has the statistical analysis been performed appropriately and rigorously? 

Reviewer #1: I Don't Know

Reviewer #2: Yes

Reviewer #3: I Don't Know

4. Have the authors made all data underlying the findings in their manuscript fully available?

Reviewer #1: Yes

Reviewer #2: (No Response)

Reviewer #3: Yes

5. Is the manuscript presented in an intelligible fashion and written in standard English?

Reviewer #1: Yes

Reviewer #2: Yes

Reviewer #3: Yes

6. Review Comments to the Author

Reviewer #1: The authors made efforts to answer to any reviewed points.

1-Concerning the first point :

Although not being a well-design randomized trial, the recruitment process has to be described and is always missing here.

Above all, it needs to be the same between populations groups (example of additional required information: inclusion, exclusion criteria etc..).

The small size of sample is the major limitation of study. The authors are aware and, unfortunately, it can’t be solved at this point of revision.

Determining the feasibility of the multiplex ECLIA assay is based on deep and multi-variable statistical analysis, which being impaired by this minimal number of person (see above) as wel as from a unrandomized population selection (‘convenience’ samples)

In addition to the heterogenicity of the sample groups, it is still not precised if the developped method interest is to the diagnostic (1) and/or the surveillance (2).

(1) In view of the current epidemiological context, diagnostic tools need to be operational to respond to emergency (outbreak), opposing the authors sentence "a rigorous case-control study was infeasible in the midst of an emergency outbreak response".

(2) In addition, a sero-surveillance tool development in "real-world" needs to be reliable, in continue, meaning to be evaluate from disease progression data. Both unfeasible in emergency context and unvalued in accordance to the illness outcome, the interest of this tool is questioned.

It is even less useful to the preparedness as authors saying being a " pre-pandemic to pandemic imperfectly comparison study. "

To conclude, the first answer remains unsatisfactory.

2-The second point is partially it. While the authors claims that the control samples were tested for COVID-19 using the same diagnostic assay as the COVID-19 patients, the "RT-PCR or RT-qPCR" molecular approach is not precised.

3- The accuracy of a clinical study is based on precise and available personal information of participations and not on approximative data " age range and sex for COVID-19 and Control subjects in Table 1 and in S1 Table". Unfortunately, the militaries, the U.S. Department of Defense civilians and contractors are far from to reflect a wide demographic range of U.S. population, including manual worker and/or low-class.

4-The fact that it is not possible to determine general trends in pre-pandemic coronavirus exposure in these individuals with diverse geographic history makes antibody responses analyses inaccuracy (specificity, sensitivity, cross specificity, immune response memory)

5-The ages have not to be approximative in the groups.

6- This point is satisfied by additional sentence

Minor points

The authors took the trouble to answer at any minor points although being aware of any study limitations: an un sufficient power to identify significant differences in serological data with respect to sex and age, a high (apparent 50%) IgM seropositivity for MERS-CoV in the Control Group and absence of diseases progression. These three points oppose to validation this tool use in the Covid-19 management and/or survey.

As being much more sensitive than a regular ELISA (we had previously published formal, extensive comparisons of the platforms), ECLIA may be attractive in the screening campaign by in particular conjugating the IgM/IgA than IgM/IgG.

Conclusion

The authors’ efforts in rewriting, in providing evidence and in arguing reviewer comments are visible. In view of preliminary results, enrich works by increasing the size of population groups, by providing the personal information and by following the diseases progression or conversely, the asymptomatic forms maintenance should be carried out.

From that, a sufficient accuracy is hoped to reach as well as to erase the study limitations and to convince to develop the platform of covid diagnostic based on ECLIA process. But, It is not the case now.

Reviewer #2: (No Response)

Reviewer #3: The manuscript entitled “Serological profiles of pan-coronavirus-specific responses in COVID-19 patients using a multiplexed electro-chemiluminescence-based testing platform” by Prof. Elke Bergmann-Leitner presented a new 10-plex electro-chemiluminescence-based assay to measure IgM and IgG responses to the spike proteins from multiple human coronaviruses including SARS-CoV-2, assess the epitope specificity of the SARS-CoV-2 antibody response against full-length spike protein, receptor-binding domain and N-terminal domain of the spike protein, and the nucleocapsid protein. The article has been carefully revised according to the reviewers’ comments and well organized. Publication in PLOS ONE is recommended after correction of some typos and grammar mistakes.

7. PLOS authors have the option to publish the peer review history of their article (what does this mean?). If published, this will include your full peer review and any attached files.

Reviewer #1: No

Reviewer #2: **Yes: **Pascal Pineau

Reviewer #3: No

---

## [Author Response · Author response to Decision Letter 1]

17 May 2021

Response to Reviewer 1:

Comment 1: Although not being a well-design randomized trial, the recruitment process has to be described and is always missing here. Above all, it needs to be the same between populations groups (example of additional required information: inclusion, exclusion criteria etc..). Determining the feasibility of the multiplex ECLIA assay is based on deep and multi-variable statistical analysis, which being impaired by this minimal number of person (see above) as wel as from a unrandomized population selection (‘convenience’).

>>This study was not intended to validate the ECLIA assay for immediate use in COVID-19 management or serosurveillance; it was intended to demonstrate its feasibility for potential use towards those applications, based on its high throughput, low blood volume requirement, and the high sensitivity and specificity we observed in this study. Furthermore, we present bioinformatics approaches to interpret the complex data sets this assay generates, resulting in a more reliable identification of infection. We respectfully disagree with the reviewer that large, randomized trial are necessary to determine the feasibility of an assay. Feasibility tests frequently involve relatively small samples of convenience to determine if an assay has potential value for further use. 

Regarding to concern whether the study is powered to yield statistically significant – the biostatistician on our team performed multiple tests (described in the Materials and Methods section) to assure that sample sizes were sufficient and no over-modeling occurred. <<

Comment 2: It is still not precised if the developped method interest is to the diagnostic (1) and/or the surveillance (2).

(1) In view of the current epidemiological context, diagnostic tools need to be operational to respond to emergency (outbreak), opposing the authors sentence "a rigorous case-control study was infeasible in the midst of an emergency outbreak response".

(2) In addition, a sero-surveillance tool development in "real-world" needs to be reliable, in continue, meaning to be evaluate from disease progression data. Both unfeasible in emergency context and unvalued in accordance to the illness outcome, the interest of this tool is questioned.

It is even less useful to the preparedness as authors saying being a " pre-pandemic to pandemic imperfectly comparison study. "

>>The study we describe was intended to validate the assay for immediate use in diagnostics or serosurveillance. Instead, it presents a case for the future use of this technology to obtain more insights (broader immune profile) than e.g., a one-parameter serum ELISA, and it describes the computational tools for reliably identifying the immune status of the serum donors. Our results justify the conduct of follow-on studies – and potentially a broader application – of the approach we describe. Regarding its ultimate application, we would like to emphasize that an assay can be of interest for both diagnostic and serosurveillance purposes and the two are not mutually exclusive. Here, we are presenting evidence that this approach would be useful for both.<<

Comment 3: While the authors claims that the control samples were tested for COVID-19 using the same diagnostic assay as the COVID-19 patients, the "RT-PCR or RT-qPCR" molecular approach is not precised.

>>We have updated the manuscript to include the details of the COVID-19 PCR test used in the study: “All COVID-19 diagnoses were confirmed using a nasopharyngeal swab and RT-PCR-based diagnostic assay (Centers for Disease Control 2019-nCoV RT-PCR diagnostic panel run on the Applied Biosystems 7500 platform)."<<

Comment 4: The accuracy of a clinical study is based on precise and available personal information of participations and not on approximative data " age range and sex for COVID-19 and Control subjects in Table 1 and in S1 Table". Unfortunately, the militaries, the U.S. Department of Defense civilians and contractors are far from to reflect a wide demographic range of U.S. population, including manual worker and/or low-class.

We are subject to stringent IRB rules that prohibit investigators from divulging exact ages of study participants as part of restrictions on releasing Personally Identifiable Information, and thus, we are not allowed to include such details in manuscripts. Age ranges to within 10 years should be sufficient for scientific purposes while maintaining the privacy of the patients and this had not been raised as a concern in previous studies (we have conducted immunoprofiling on human samples from a variety of clinical trials). We strongly disagree with the reviewer that the donors of the samples used for this study are somehow skewed and are ethnically or socioeconomically insufficiently diverse. Our samples were obtained from individuals with very different jobs within the DoD (and, thus, very different educational, income, and socioeconomic status). While we may not have included very low-income/inner-city participants or a homeless cohort (similar to many other clinical trials), we are not entirely sure how including such additional populations would have changed the conclusions of the study.<<

Comment 5: The fact that it is not possible to determine general trends in pre-pandemic coronavirus exposure in these individuals with diverse geographic history makes antibody responses analyses inaccuracy (specificity, sensitivity, cross specificity, immune response memory)

>>We acknowledged the limitations of using pre-pandemic samples in our analyses and this limitation applies to virtually all published COVID-19 serology studies. However, this is precisely why it is essential to refine serological tests for SARS-CoV-2, both in terms of sensitivity and specificity, which is what we describe in this study. <<

---

## [Editor Report · Decision Letter 2]

19 May 2021

Serological profiles of pan-coronavirus-specific responses in COVID-19 patients using a multiplexed electro-chemiluminescence-based testing platform

PONE-D-21-00516R2

Dear Dr. Bergmann-Leitner,

We’re pleased to inform you that your manuscript has been judged scientifically suitable for publication and will be formally accepted for publication once it meets all outstanding technical requirements.

Kind regards,

Etsuro Ito

Academic Editor

PLOS ONE

---

## [Editor Report · Acceptance letter]

24 May 2021

PONE-D-21-00516R2 

Serological profiles of pan-coronavirus-specific responses in COVID-19 patients using a multiplexed electro-chemiluminescence-based testing platform 

Dear Dr. Bergmann-Leitner:

I'm pleased to inform you that your manuscript has been deemed suitable for publication in PLOS ONE. Congratulations! Your manuscript is now with our production department. 

Kind regards, 

on behalf of

Prof. Etsuro Ito 

Academic Editor

PLOS ONE